# Hydrolysis of Methylumbeliferyl Substrate Proxies for Esterase Activities as Indicator for Microbial Oil Degradation in the Ocean: Evidence from Observations in the Aftermath of the Deepwater Horizon Oil Spill (Gulf of Mexico)

Kai Ziervogel [1],* , Manoj Kamalanathan [2],* and Antonietta Quigg [2]

[1] Institute for the Study of Earth, Oceans and Space, University of New Hampshire, Durham, NH 03824, USA
[2] Department of Marine Biology, Texas A&M University at Galveston, Galveston, TX 77553, USA; quigga@tamug.edu
* Correspondence: kai.ziervogel@unh.edu (K.Z.); manojka@tamug.edu (M.K.)

**Abstract:** Biological oil weathering facilitated by specialized heterotrophic microbial communities plays a key role in the fate of petroleum hydrocarbon in the ocean. The most common methods of assessing oil biodegradation involve (i) measuring changes in the composition and concentration of oil over time and/or (ii) biological incubations with stable or radio-labelled substrates. Both methods provide robust and invaluable information on hydrocarbon biodegradation pathways; however, they also require extensive sample processing and are expensive in nature. More convenient ways to assess activities within microbial oil degradation networks involve measuring extracellular enzyme activity. This perspective article synthesizes previously published results from studies conducted in the aftermath of the 2010 Deepwater Horizon (DwH) oil spill in the northern Gulf of Mexico (nGoM), to test the hypothesis that fluorescence assays of esterases, including lipase activity, are sensitive indicators for microbial oil degradation in the ocean. In agreement with the rates and patterns of enzyme activity in oil-contaminated seawater and sediments in the nGoM, we found close correlations between esterase activity measured by means of methylumbeliferyl (MUF) oleate and MUF butyrate hydrolysis, and the concentration of petroleum hydrocarbons in two separate laboratory incubations using surface (<1 m) and deep nGoM waters (>1200 m). Correlations between esterase activities and oil were driven by the presence of chemical dispersants, suggesting a connection to the degree of oil dissolution in the medium. Our results clearly show that esterase activities measured with fluorogenic substrate proxies are a good indicator for oil biodegradation in the ocean; however, there are certain factors as discussed in this study that need to be taken into consideration while utilizing this approach.

**Keywords:** microbial oil degradation; esterase; MUF butyrate; MUF oleate; petroleum hydrocarbons; dispersant; Deepwater Horizon oil spill; Gulf of Mexico

## 1. Background

Crude oils are complex mixtures of organic compounds that can contain upwards of 17,000 distinct chemical components [1]. Within this complexity, crude oil components can be classified into four operationally defined groups of chemicals: saturated hydrocarbons (e.g., linear, branched, and cyclic alkanes), aromatic hydrocarbons, and the more polar, highly complex and dense resins and asphaltenes. Light oils are typically high in saturated and aromatic hydrocarbons, with a smaller proportion of resins and asphaltenes. Heavy oils have a much lower content of saturated and aromatic hydrocarbons and a higher proportion of the more polar chemicals [2].

Crude oil is introduced into the oceans through various sources, including natural seeps as well as pipeline and tanker leaks and spills. Its fate is greatly determined by

the concerted efforts of hydrocarbon-degrading microbes whose rates and activities are of key interest to scientists and first responders to predict the fate of spilled oil in the environment. The largest open ocean oil spill to date in the United States occurred in 2010 in the northern Gulf of Mexico (nGoM), when the sunken Deepwater Horizon (DwH) oil platform accidentally discharged approximately five million barrels of light Louisiana crude oil and at least 250,000 metric tons of natural gas to deep GoM waters over a period of about 3 months [3]. Massive quantities of the discharged gas and dissolved hydrocarbons were entrained in oil plumes at depths between 1000 m and 1300 m [4,5]. Shortly after the spill, ~2.9 million liters of the dispersing agent Corexit was applied [6]. Microbial communities throughout the water column, including the deep-water oil plumes, responded to this unprecedented accident with the rapid oxidation rates of dissolved gases and hydrocarbons assessed by means of $^{13}$C-labeled propane and ethane conversion into $^{13}$C-CO$_2$ and tritiated methane conversion into $^3$H$_2$O [7,8], respectively. Furthermore, oil degradation rates were quantified from changes of hydrocarbon concentrations (alkanes in deep water [9]), microbial cell numbers and respiration rates [10], as well as nutrient anomalies in the water column [11]. For a more detailed discussion of the microbial responses and activities during the DwH and other spills, we refer to the many review articles on this subject [2,12–14].

An alternative approach to the above-mentioned methods involves measuring the activities of hydrolytic enzymes, such as esterases, which are involved in the breakdown of hydrocarbon degradation products. In general, the microbial breakdown of hydrocarbons, such as alkanes, is a well characterized process that occurs at the outer cell membrane as well as in the intracellular space. Alkane degradation is initiated by the hydroxylation of alkanes at either the terminal or subterminal carbon, followed by further oxidation to the carboxylic acid (terminal hydroxylation), or to acetate plus a carboxylic acid that is shorter by two carbons (subterminal hydroxylation); the latter involves an ester intermediate that is hydrolyzed by esterases [15]. Extracellular esterases, including lipases (i.e., a subclass of esterases), are also involved in the breakdown of hydrocarbon byproducts formed from photooxidation [16] and from reactions with reactive oxygen species in seawater [17,18]. The extracellular transformation of hydrocarbons by laccases and peroxidases from marine fungi have also been found to generate esterified molecules that act as substrates for lipases [19,20].

The role of extracellular lipases in transforming and emulsifying hydrocarbons has previously been demonstrated in several culture studies with known hydrocarbon degraders [21,22]. Close correlations between lipase activities and hydrocarbons were also found in a laboratory study with oil-contaminated soils, supporting the notion that lipases play a key role in oil biodegradation [23]. This perspective article focuses on esterases as bioindicators for oil degradation in the ocean. Hereafter, we use esterases as a general term for hydrolytic enzymes that cleave ester bonds, which include lipases.

The focus on esterases stems from the fact that they can be monitored in environmental samples using well-established and easy-to-use bioassays, making them applicable to a broad community of oil remediation researchers. Enzyme assays target the evolution of hydrolysis products of specific substrates that are added to the experimental sample, yielding potential rates of naturally occurring enzymes. A more holistic view of enzymatic machineries requires the application of genomic techniques, such as transcriptomics, proteomics, and metabolomics, which reveal invaluable insights into gene expression and thus the metabolic functions of microbial communities present in the sample. Expression data from -omics, however, do not directly correlate with enzymatic rates.

Examples for bioassays on esterase activities in microbial oil degradation networks include those using b-naphtyl derivates as substrates proxies [22]. Others measured the accumulation of butyric acid in experimental soils as a hydrolysis product of tributyrin [23], i.e., a common substrate for esterase-producing bacteria [24]. A more sensitive enzyme assay compared with the former is based on the fluorescence detection of hydrolysis products of extracellular esterases, involving the use of fluorogenic substrate proxies that typically con-

sist of a monomer (e.g., fatty acids) linked to a fluorescent tag (methylumbelliferyl—MUF). MUF substrates for esterases (and other common hydrolases) are commercially available (e.g., Millipore Sigma, St. Louis, MO, USA; Table 1) and their application to assay naturally occurring enzyme activities in aquatic environments is well established [25]. MUF assays are based on the enzymatic cleavage of the fluorophore from the substrate complex, which results in an increase in fluorescence in the incubation medium. Given that MUF substrates are restricted from transport into the cells' interior [26], the assays target extracellular enzymes that are located in the periplasmic space, on the outer cell wall, and/or freely dissolved in the ambient water [27].

**Table 1.** List of MUF substrates that have been used to assay esterase activity in the ocean.

| Substrate | Empirical Formula | Millipore Sigma Product Number |
|---|---|---|
| 4-Methylumbelliferyl butyrate (MUF BU) | $C_{14}H_{14}O_4$ | 19362 |
| 4-Methylumbelliferyl stearate (MUF ST) | $C_{28}H_{42}O_4$ | M1010 |
| 4-Methylumbelliferyl palmitate (MUF PA) | $C_{26}H_{38}O_4$ | M7259 |
| 4-Methylumbelliferyl oleate (MUF OL) | $C_{28}H_{40}O_4$ | 75164 |

MUF substrates are commonly used in the ocean to measure the activities of the extracellular enzymes involved with the breakdown of freshly produced organic matter (e.g., α- and β-glucosidase cleaving MUF-α-D-glucopyranoside and -β-D-glucopyranoside, respectively) and the recycling of inorganic nutrients (e.g., alkaline phosphatases cleaving MUF phosphate) [28–30]. Despite their importance for organic matter cycling—particularly for less labile fractions of the organic matter pool [31]—comparatively fewer studies have applied MUF substrates to investigate esterase activities in the ocean. These studies mainly focused on enzyme activities in the deep-sea sediments [32–36] and tidal flats [37] (Table 2).

**Table 2.** Previous published results on esterase activities in the ocean.

| Sampling Location; Type of Sample; Sampling Depth | Substrates Assayed | Comments | References |
|---|---|---|---|
| NW Atlantic; Intertidal sediments (Gulf of Maine) | MUF ST, MUF PA | MUF ST hydrolysis not detectable; MUF PA hydrolysis rates up to one order of magnitude lower than glucosidases. | [37] |
| NE Atlantic; deep-sea sediments (4500 m) | MUF ST | MUF ST hydrolysis in the same range as glucosidases, and not stimulated by fresh DOM. | [32,36] |
| Arabian Sea; deep-sea sediments (3000–4500 m) | MUF BU, MUF ST | MUF ST hydrolysis at detection limit and one order of magnitude lower than MUF BU. | [33] |
| Northern Gulf of Mexio; surface water near DwH spill site | MUF BU | MUF BU hydrolysis up to four times higher in oil-contaminated compared with uncontaminated surface waters. MUF BU two orders of magnitude higher than phosphatases. | [10] |
| | MUF BU | MUF BU hydrolysis in MOS double that of glucosidases. | [38] |
| Northern Gulf of Mexico; deep-water oil plume | MUF BU | MUF BU hydrolysis 1.75 times higher in plume compared with non-plume water and one order of magnitude higher than glucosidases. | Ziervogel (unpubl.) |
| Northern Gulf of Mexico, deep-sea sediments affected by the DwH fallout (1500–1900 m) | MUF BU | MUF BU hydrolysis one to two orders of magnitude higher than glucosidases. | [39] |
| Northern Gulf of Mexico; resuspended deep-sea sediments near natural seeps (530–1600 m) | MUF BU | MUF BU hydrolysis one to two orders of magnitude higher compared with glucosidases. | [40] |

Insights from esterase activities within the microbial oil degradation cascade come from studies conducted shortly after the onset of the DwH spill in the nGoM. For instance, elevated esterase activities were found in oil-contaminated surface waters relative to uncontaminated waters near the DwH site [10]; in oily organic matter aggregates known as marine oil snow (MOS) [38] that formed ex situ in the DwH spill site; and in the deep-water oil plume relative to non-plume deep waters (Table 2). Moreover, elevated esterase activities were found in deep-sea sediments affected by the DwH oil fallout [39], and in a more recent study conducted with nGoM sediments taken near natural oil and gas seeps [40]. These results indicate a connection between extracellular esterase activities and the cycling of petroleum hydrocarbons, which, to the best of our knowledge, has not yet been established for marine environments. This perspective paper synthesizes previously published results on esterase activities measured in oil biodegradation experiments to test the following hypothesis:

**Hypothesis 1.** *Esterase activities assayed with MUF substrate proxies are suitable indicators for microbial oil degradation in the ocean.*

## 2. Esterase Activities in Oil-Contaminated Water under Controlled Laboratory Conditions

Following the DwH oil spill, several research projects funded by the Gulf of Mexico Research Initiative (GOMRI) were initiated to study the effects of oil contamination on nGoM ecosystems. Among those were our two research consortia (ECOGIG—ECosystem responses to Oil and Gas Inputs into the Gulf, and ADDOMEx—Aggregation and Degradation of Dispersants and Oil by Microbial Exopolymers) that mainly focused on microbial community responses to and interactions with crude oil and chemically dispersed oil (Corexit). This work involved a number of laboratory incubations and mesocosm experiments over time scales ranging from days to months that resulted in numerous publications [38,41–46]. The present article synthesizes the results of two of these laboratory experiments in which esterase activities were measured as supporting data for microbial metabolic rates in oil-contaminated waters. To test our hypothesis, we emphasize the observed correlations between esterase activities and oil concentrations in the two case studies outlined below.

### 2.1. Case Study 1: Surface Water Incubation

Esterase activities in oil-contaminated surface waters were investigated as a part of a large scale mesocosm experiment with waters collected from the nGoM (29°16′ N 94°48′ W) in May/June 2017. The overarching goal of the study was to assess dynamics of microbial exopolymeric substances (EPS) in the presence of Macondo surrogate oil and dispersant (Corexit). For a detailed description of the experimental set up, oil preparation, and sample treatment and analysis, see [41]. In brief, a total of 9 mesocosm tanks (each holding 87 L of oil-amended and unamended seawater) were incubated for 15 days using a 12:12 day/night cycle and an ambient temperature (21 °C). Treatments included a control, a water accommodated fraction (WAF), and a diluted chemically enhanced WAF (DCEWAF). Esterase activities were measured once a day throughout the incubation by means of MUF oleate (MUF OL) hydrolysis, as described in [47,48]. Changes in WAF and DCEWAF over time were monitored by measuring the fluorescence of dichloromethane water extracts at 260/358 nm excitation/emission (i.e., estimated oil equivalents—EOE [49]). EOE has been shown to be a reliable fluorescent estimate of oil concentration in seawater [50]. Hydrocarbon extraction from each of the mesocosm tanks was performed as described in [41]. N-alkanes and PAHs were measured using an Agilent 6890 gas chromatograph with a mass selective detector and an Agilent 5890 gas chromatograph with a flame ionization detector. Total hydrocarbons were defined as the sum of resolved peaks (n-alkanes, branched hydrocarbons, PAH, and nitrogen- and sulfur-containing compounds), and the unresolved complex mixture (UCM). The initial levels of total hydrocarbons were about 1910 $\mu g \, L^{-1}$.

In the first part of the experiment (0—72 h), esterase activities were significantly higher in the DCEWAF compared to unamended surface water, which served as the non-oil control treatment ($p = 0.0118$, paired t-test; Figure 1A). Esterase activities in the WAF treatment showed a delayed induction compared to DCEWAF with higher than the control values observed between 48 and 144 h ($p = 0.0269$, paired t-test). Esterase activities in the WAF and DCEWAF showed significant correlations with EOE (Figure 1B); however, esterase activities and the total hydrocarbon concentration showed a significant linear relation in DCEWAF, but not in the WAF treatment, considering a 5% significant level (Figure 1C). Such treatment dependent responses were also observed when the esterase activities were compared to the n-C17/Pristane, n-C18/Phytane, and n-C17/n-C18 ratios (Table 3), which can serve as indicators of oil biodegradation [51]. In the DCEWAF, a significant correlation was observed between esterase activities and n-C17/Pristane, and n-C18/Phytane ratios, but not n-C17/n-C18. However, in the WAF treatment, a significant correlation was only observed between esterase activities and n-C17/n-C18 ratios.

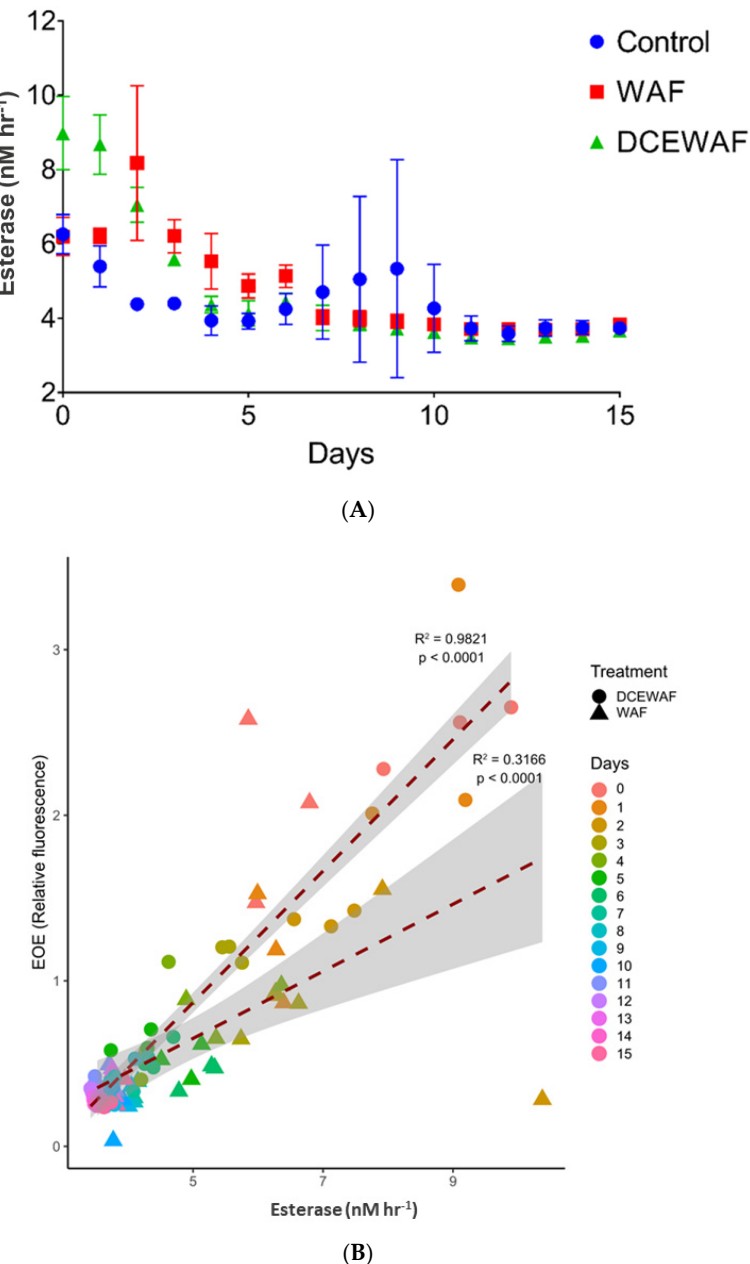

**Figure 1.** *Cont.*

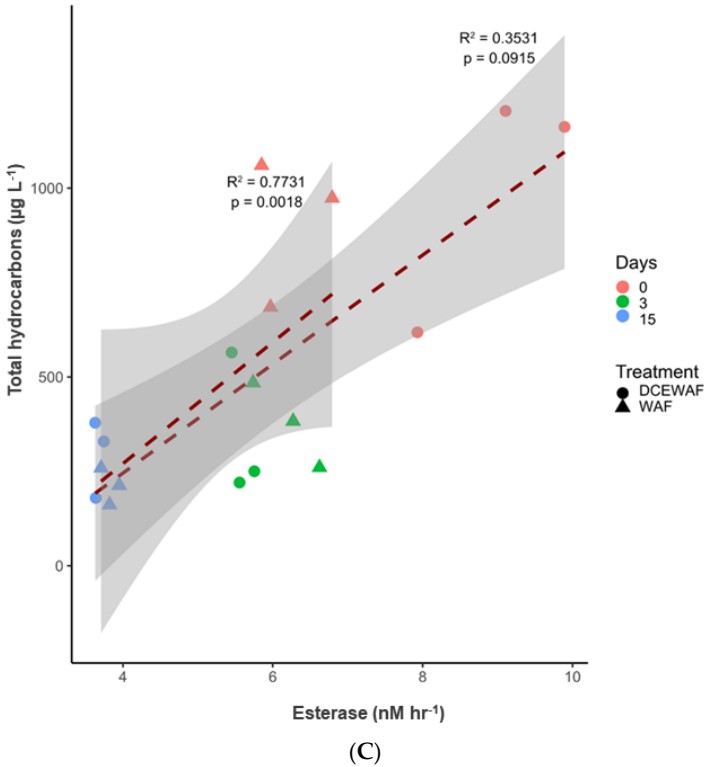

(**C**)

**Figure 1.** Esterase activities measured by means of MUF oleate hydrolysis and oil concentrations in the surface water incubation (case study 1). (**A**): Time courses of esterases (average rates of *n* = 3; ± standard deviations) in the two oil treatments (WAF and DCEWAF) and the non-oil control water. Note that the elevated enzyme activities in DCEWAF at day 0 could have been induced during the overnight treatment of the amendment (see [41]). (**B**): Correlation between esterases and estimated oil equivalents (EOE). (**C**): Correlations between esterases and total hydrocarbons. Plot B and C show hydrolysis rates from the three replicates (note the outlier in esterase activity of one of the replicates at day 2 in the WAF, B). Esterase activities and EOE are replotted from [48]; total hydrocarbon concentrations are from GRIDCII https://data.gulfresearchinitiative.org/data/R4.x263.000:0053 (accessed on 11 April 2022).

**Table 3.** Pearson correlation coefficient ($r^2$) and *p*-values for surface water activities of esterase and oil biodegradation indicator ratios.

|  | n-C17/Pristane | n-C18/Phytane | n-C17/n-C18 |
|---|---|---|---|
| WAF | $r^2 = 0.4365$ ($p = 0.053$) | $r^2 = 0.4115$ ($p = 0.063$) | $r^2 = 0.6953$ ($p = 0.005$) |
| DCEWAF | $r^2 = 0.784$ ($p = 0.001$) | $r^2 = 0.8808$ ($p < 0.001$) | $r^2 = 0.265$ ($p = 0.156$) |

*2.2. Case Study 2: Deep-Water Incubation*

Esterase activities in deep nGoM waters were measured in a laboratory experiment designed to investigate the effects of oil and Corexit exposure on deep water bacterial communities. Water for this experiment was collected at a depth of ~1200 m near a natural oil seep (27°21′ N, 90°34′ W) in March 2013. A detailed description of the experimental set-up, oil preparation, and sample treatment and analysis is given in [45]. In brief, a total of 72 experimental bottles (2-L Pyrex bottles) were filled with oil amended and unamended deep water (the latter served as non-oil controls). As in case study 1, oil and Corexit were added to the experimental bottles as WAF and CEWAF. The bottles were incubated in the dark on a roller table at 7 °C, which was the in situ temperature at the time of sampling. Esterase activities were measured before the initiation of the incubation (day 0), and after

7, 16, 28, and 42 days by means of MUF butyrate (MUF BU) hydrolysis following the same procedure as described above [47]. Hydrocarbon analysis was conducted using a Gas Chromatography/Mass Selective Detector (GC/MSD; Agilent 7890GC/5975MSD) as described in [52]. Compound identification was based on individual mass spectra and retention times in comparison to library data and to authentic standards that were injected and analyzed under the same conditions. The following compounds were identified: *n*-Alkanes, Hexadecane, Naphthalene, and Phenanthrene. The sum of the latter four compounds plus unresolved peaks (UCM) was expressed as total hydrocarbons. The initial levels of total hydrocarbons were about 145 $\mu g\,L^{-1}$ and thus in the same range than those found in the deep oil plumes during the DwH spill [45].

Deep water esterase activities increased throughout the time course of the WAF incubation, reaching one and two orders of magnitude higher levels compared with the CEWAF ($p = 0.0416$, paired t-test) and the control treatment ($p = 0.0138$, paired t-test), respectively (Figure 2A). Esterase activities in both oil treatments showed significant correlations with total hydrocarbon concentrations, although correlations in the WAF were stronger than in the CEWAF (Figure 2B), following the treatment-dependent responses found in the surface water experiment. Furthermore, deep water esterase activities showed inverse correlations with the four hydrocarbon fractions analyzed in this work (Table 4). Correlations were found to be significant in the CEWAF treatment only for Hexadecane, *n*-Alkanes, and Phenanthrene.

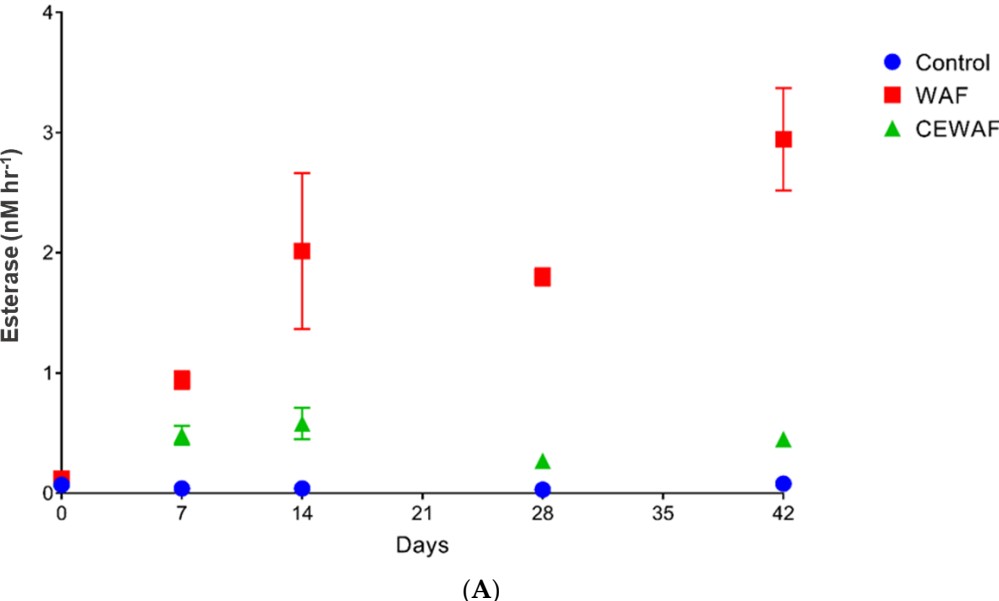

(**A**)

**Figure 2.** *Cont.*

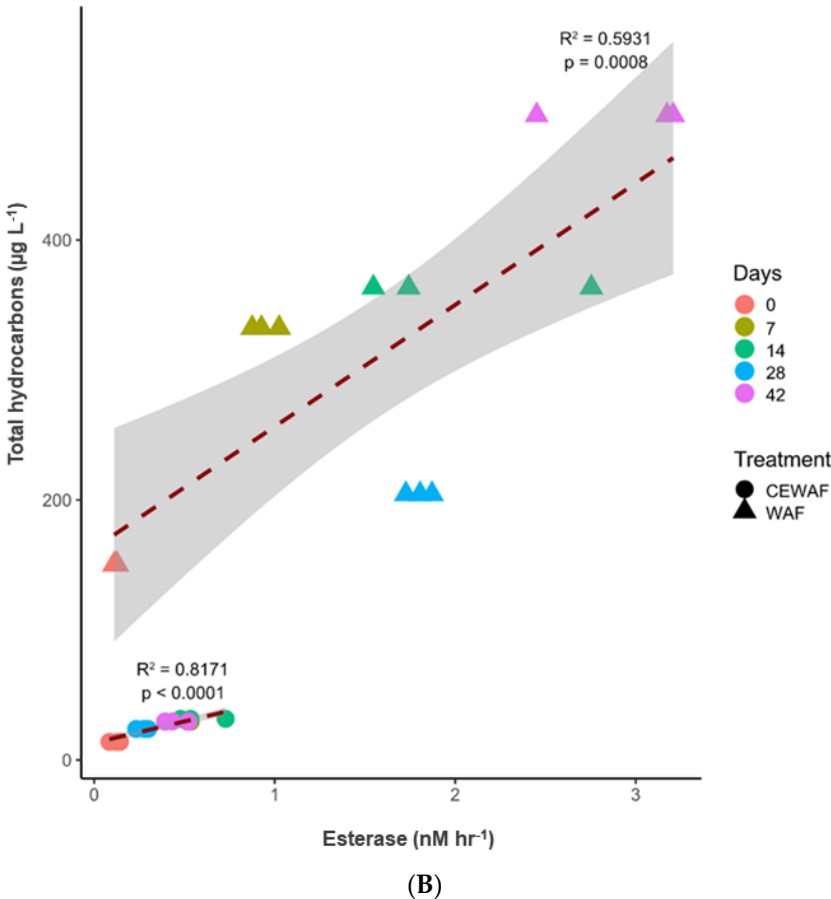

(**B**)

**Figure 2.** Esterase activities measured by means of MUF butyrate hydrolysis and oil concentrations in the deep-water incubation (case study 2). (**A**): Time courses of esterases (average rates of $n = 3$; ± standard deviations) in the two oil treatments (WAF, CEWAF) and the non-oil control water. (**B**): Correlation between lipases and total hydrocarbons. Note that hydrolysis rates of the three replicates were plotted in B. All of the data in this figure are replotted from [45].

**Table 4.** Pearson correlation coefficient ($r^2$) and *p*-values for deep water lipase activities and hydrocarbons.

|  | Hexadecane | *n*-Alkanes | Naphthalene | Phenanthrene |
|---|---|---|---|---|
| WAF | −0.83 ($p = 0.08$) | −0.85 ($p = 0.07$) | −0.87 ($p = 0.06$) | −0.88 ($p = 0.05$) |
| CEWAF | −0.94 ($p = 0.02$) | −0.89 ($p = 0.04$) | −0.85 ($p = 0.07$) | −0.92 ($p = 0.03$) |

## 3. Esterase Activities as Indicator for Oil Biodegradation

Esterase activities in our two case studies closely followed changes in oil concentrations (EOE—surface; total hydrocarbons—surface and deep), indicating that esterases play an important role in the microbial degradation of oil in the ocean. Our results support previous findings from culture experiments on microbial degradation of alkanes [22] and biodegradation of oil in soils [23]. Notable insights into these processes arose from our investigation of esterase activities in the experiments with chemically enhanced oil (CEWAF/DCEWAF), where the correlations between esterase activities and total hydrocarbons were generally stronger than in the WAF only treatments in both experiments. Moreover, esterase activities in the DCEWAF surface water incubation showed significant correlations with the three hydrocarbon ratios associated with oil biodegradation, while only two of these ratios significantly correlated with esterase in the WAF (Table 3). Such treatment-dependent responses could have been due to the strong influence of dispersant

on the dissolution of oil components and hence their bioavailability. Previous studies have shown that the addition of dispersant can enhance the dissolution of alkanes and PAHs and therefore aid in their increased degradation [41,53,54]. Byproducts from the primary degradation of dissolved alkanes and PAHs provide suitable substrates for esterases [55,56]. Ester bonds within the Corexit complex itself may also be targets for extracellular esterases, thus stimulating their activities [10,45]. When considering EOE as a measure of oil concentrations, as in the case of the surface water experiment, enzymatic responses to the addition of Corexit were less pronounced as both DCEWAF and WAF showed similarly strong correlations with esterases (Figure 1B). This pattern could have been due to the fact that EOE mainly measures aromatics and not saturated hydrocarbons (alkanes) with dispersants enhancing the accommodation of the former rather than the latter [57,58].

The structure and metabolic functions of oil-degrading microbial communities have been shown to be affected in the presence of Corexit in the two laboratory incubations presented here [43,45], and other microbial oil degradation experiments [59]. As in the case of our deep-water incubation, the presence of dispersant (CEWAF) selected for potential dispersant-degrading bacteria (*Colwellia* sp.), which also bloomed in situ in nGoM deep waters during the DwH spill [45]. In contrast, the WAF amendment stimulated the growth of natural hydrocarbon-degrading bacteria of the genus *Marinobacter*. As a consequence, the hydrocarbon degradation rates presented in Kleindienst et al. [45] were lower in the CEWAF compared with the WAF treatment. This pattern explains the higher esterase activities in the latter compared with the former treatment (Figure 2A,B), underlining the role of esterases within the oil biodegradation cascade.

Another notable observation between the surface and the deep-water incubation was the difference in the time courses of esterase activities. While peak levels of oil and esterases in experimental surface waters were found at the beginning of the incubation, the opposite trend was observed in the deep-water experiment where total hydrocarbons and esterase activities increased during the initial phase (CEWAF) and throughout the 42-day incubation (WAF). Margesin et al. [23] also reported increasing levels of esterases during their weeks-long biodegradation experiment of diesel oil in soils, supporting the notion that esterases are involved in the breakdown of hydrocarbon degradation byproducts that accumulate over the time course of oil biodegradation [60]. The latter may have caused the negative correlation between esterase activities and hydrocarbon concentrations in the deep-water incubation (Table 4).

## 4. Conclusions

Our results demonstrate that the activities of extracellular esterases measured by means of MUF substrates can be used as an indicator of oil biodegradation in the ocean. We note, however, that there are several caveats to be considered when applying this method. First, the paucity of literature data on esterase activities in the ocean and, in particular, oil-contaminated seawater complicates the comparison among different marine environments and oil spill scenarios and, thus, the generalization of the findings from the nGoM. Second, we still lack an in-depth understanding of the enzymatic pathways that involve extracellular esterases within microbial oil degradation networks, particularly in the presence of chemical dispersants, such as Corexit. Finally, the use of MUF substrates in oil degradation assays involve uncertainties as a broad range of marine microbes, including those not directly involved in oil degradation, produce enzymes capable of hydrolyzing fluorogenic substrates, such as MUF oleate and MUF butyrate. The hydrolysis rates of these substrates in non-oil control treatments can give insights into the 'natural background' of hydrolysis rates and are therefore important when considering the use of esterases as predictors of oil biodegradation. Bioassays with MUF substrates may be linked with transcriptomic data in future studies to gain further insights into the role of esterases in oil biodegradation processes in the ocean.

**Author Contributions:** Conceptualization, K.Z., M.K. and A.Q.; methodology, K.Z. and M.K.; data curation, K.Z. and M.K.; writing—original draft preparation, K.Z. and M.K.; writing—review and

editing, A.Q.; visualization, M.K.; funding acquisition, K.Z. and A.Q. All authors have read and agreed to the published version of the manuscript.

**Funding:** This research was funded by the Gulf of Mexico Research Initiative (GOMRI) supporting the ECOGIG (K.Z.) and ADDOMEx (M.K., A.Q.) consortia. This is ECOGIG contribution #589.

**Informed Consent Statement:** Not applicable.

**Data Availability Statement:** The data presented in this study are publicly available through the Gulf of Mexico Research Initiative Information & Data Cooperative (GRIIDC) at https://data.gulfresearchinitiative.org/ (accessed on 11 April 2022).

**Acknowledgments:** The authors would like to thank Shawn Doyle and Jason Sylvan (both Texas A&M University) for helpful discussions and text edits.

**Conflicts of Interest:** The authors declare no conflict of interest.

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
