# Peer review of "Hydrolysis of Methylumbeliferyl Substrate Proxies for Esterase Activities as Indicator for Microbial Oil Degradation in the Ocean: Evidence from Observations in the Aftermath of the Deepwater Horizon Oil Spill (Gulf of Mexico)"

_jmse, doi:10.3390/jmse10050583_

Round 1

Reviewer 1 Report

The data presented in this article provide some support for the authors thesis that their fluorescence-based assay of esterase activity may be a useful approach for monitoring biodegradation rates of petroleum products spilled into receiving waters.  However, there are serious flaws in the data used, and the analysis and interpretations of the data presented, that along with results that partially conflict, preclude acceptance. 

Results of field observations presented in the first part of the article provide suggest a possible association between the esterase assay and increased microbial degradation of petroleum products, and there is a plausible mechanistic linkage between molecular biodegradation pathways of normal alkanes and esterases, which encourage more rigorous investigation.  The two mesocosm studies that are the focus of the remainder of the article provide some additional support, but interpretation of the results presented is highly problematic on several counts.

The primary objectives of the two mesocosm experiments was to estimate microbial degradation rates in surface and in deep marine waters.  These two experiments were conducted independently, at different locations and times, by different research groups that often used different methods.  Rigorous comparison of results in such situations requires considerable care regarding experimental and details, especially when selectively extracting data for evaluation of hypotheses that are different from those of the studies involved, as is the case here.  Unfortunately, inappropriate selection of data and important methodological differences in the data compared seriously impair the validity of the arguments presented in this article.

In the first mesocosm study, the actual chemistry analysis measurements are inaccurately portrayed.  The actual total petroleum hydrocarbon (TPH) measurements were based on GC-FID results for the sum of resolved peaks (which includes the n-alkanes along with a great many branched hydrocarbons, PAH, and nitrogen- and sulfur-containing compounds), and the UCM (unresolved complex mixture).  This is not quite the same as "...the sum of n-alkanes and PAHs (Naphthalene, C1-Naphthalenes, C2-Naphthalenes, C3-Naphthalenes, C4-Naphthalenes) plus unresolved peaks (unresolved complex matter – UCM).", as stated in the submitted article text. Coefficients of determination (misrepresented as "Pearson correlation coefficients" in the text) between esterase activity and ratios of C17/pristane and C18/phytane are appropriate given the mechanistic role of esterases in n-alkane degradation.  However, given the scant mechanistic basis for a similar correlation with the C17/C18 ratio (read carefully sec. 3.3.2 of ref. 51), the high correlation presented especially for the WAF treatment suggest either a spurious result, or perhaps a result of co-variation with other unidentified factors.  Speaking of which, consideration of any such factors is almost completely absent.

In the second mesocosm study, the results presented also suggest the possibility of spurious correlations or co-variation with factors other than those evaluated.  It is not obvious why esterase activity should be correlated with naphthalene or phenanthrene, other than through co-variation of these compounds with n-alkanes and hexadecane. Yet the similar correlations between esterase activity across n-alkanes, hexadecane, naphthalene and phenanthrene, in both the WAF and CEWAF treatments, suggests that co-variation with un-identified factors may be responsible, but this possibility is left almost entirely unconsidered.  More alarmingly, all the correlation coefficients presented in Table 4 are negative, in direct contradiction with the positive association between esterase activity and "total hydrocarbons" in Fig. 2B.  Most alarmingly of all, Fig. 2 B depicts total hydrocarbon concentrations that increase by a factor of nearly 3 over the 42 days of the experiment, but no explanation or even consideration of this very startling result is given.

From all of these issues, I cannot help but suspect that the data analysis is fundamentally flawed - certainly enough so that the credibility of the results is highly questionable.  There may be something to the use of esterase activity as an indicator of petroleum biodegradation, but this article falls far short of establishing reasonable plausibility. 

Author Response

Responses to reviewer 1:

In the first mesocosm study, the actual chemistry analysis measurements are inaccurately portrayed.  The actual total petroleum hydrocarbon (TPH) measurements were based on GC-FID results for the sum of resolved peaks (which includes the n-alkanes along with a great many branched hydrocarbons, PAH, and nitrogen- and sulfur-containing compounds), and the UCM (unresolved complex mixture).  This is not quite the same as "...the sum of n-alkanes and PAHs (Naphthalene, C1-Naphthalenes, C2-Naphthalenes, C3-Naphthalenes, C4-Naphthalenes) plus unresolved peaks (unresolved complex matter – UCM).", as stated in the submitted article text.

Response: We changed the sentence as suggested.

Coefficients of determination (misrepresented as "Pearson correlation coefficients" in the text) between esterase activity and ratios of C17/pristane and C18/phytane are appropriate given the mechanistic role of esterases in n-alkane degradation.  However, given the scant mechanistic basis for a similar correlation with the C17/C18 ratio (read carefully sec. 3.3.2 of ref. 51), the high correlation presented especially for the WAF treatment suggest either a spurious result, or perhaps a result of co-variation with other unidentified factors.  Speaking of which, consideration of any such factors is almost completely absent.

Response: We are not entirely sure what the reviewer is suggesting. It is certainly possible that unidentified factors with respect to oil composition and state of degradation may have affected enzymatic patterns in our mesocosms but it is beyond the scope of this manuscript to discuss all the possible geochemical factors that may have driven the observed enzyme activities.   

In the second mesocosm study, the results presented also suggest the possibility of spurious correlations or co-variation with factors other than those evaluated. It is not obvious why esterase activity should be correlated with naphthalene or phenanthrene, other than through co-variation of these compounds with n-alkanes and hexadecane. Yet the similar correlations between esterase activity across n-alkanes, hexadecane, naphthalene and phenanthrene, in both the WAF and CEWAF treatments, suggests that co-variation with un-identified factors may be responsible, but this possibility is left almost entirely unconsidered.  More alarmingly, all the correlation coefficients presented in Table 4 are negative, in direct contradiction with the positive association between esterase activity and "total hydrocarbons" in Fig. 2B.  Most alarmingly of all, Fig. 2 B depicts total hydrocarbon concentrations that increase by a factor of nearly 3 over the 42 days of the experiment, but no explanation or even consideration of this very startling result is given.

Response: We hypothesize in the text that esterases are involved in the breakdown of hydrocarbon degradation byproducts that accumulate over the time course of oil biodegradation as shown in other studies (Margesin et al. [23]) (l. 271-276). This may have caused (1) the observed negative correlations between esterases and HCs (Table 4) and (2) the observed increase in total HCs that included (unidentified) degradation byproducts, throughout the time course of the deep water incubation (Figure 2b).

Author Response

Responses to reviewer 2

line 122: "sedimentary enzyme activities" should be changed to "enzymic activities of sediments".

Response: We changed the sentence to ‘enzyme activities in dee-sea sediments’

lines 175, 215 : Unify the unit of the concentration to μg L-1.

line 216: DWH to DwH

line 262: "genius" to "genus".

Figure 1C and Figure 2B: Unify the unit of the concentration to μg L-1

Response: We fixed all the units and typos in the text and the figures as suggested.

Reviewer 3 Report

  1. The experiment data was little, not enough for an paper.
  2. The figure did not conform to the requirement.
  3. There is a little about the mechanism or deeper analysis of degradation effect in different environments.
  4. Some references cited were too old. 

Author Response

Responses to reviewer 3:

The experiment data was little, not enough for a paper.

Response: We cannot change the quantity of the data.

The figure did not conform to the requirement.

Response: To the best of our knowledge there are no specific requirements on figures.

There is a little about the mechanism or deeper analysis of degradation effect in different environments.

Response: This is beyond the scope of the paper.

Some references cited were too old.

Response: This is a rather subjective comment that we choose to politely ignore.

Round 2

Reviewer 3 Report

No comments and suggestions.